# FETAL-GAUGE: A Benchmark for Assessing Vision-Language Models in Fetal Ultrasound

**Hussain Alasmawi** [*]**, Numan Saeed & Mohammad Yaqub**
Mohamed bin Zayed University of Artificial Intelligence (MBZUAI), UAE

## Abstract

The growing demand for prenatal ultrasound imaging has intensified a global shortage of trained sonographers, creating barriers to essential fetal health monitoring. Deep learning has the potential to enhance sonographers' efficiency and support the training of new practitioners. Vision-Language Models (VLMs) are particularly promising for ultrasound interpretation, as they can jointly process images and text to perform multiple clinical tasks within a single framework. However, despite the expansion of VLMs, no standardized benchmark exists to evaluate their performance in fetal ultrasound imaging. This gap is primarily due to the modality's challenging nature, operator dependency, and the limited public availability of datasets. To address this gap, we present Fetal-Gauge, the first and largest visual question answering benchmark specifically designed to evaluate VLMs across various fetal ultrasound tasks. Our benchmark comprises over 42,000 images and 93,000 question-answer pairs, spanning anatomical plane identification, visual grounding of anatomical structures, fetal orientation assessment, clinical view conformity, and clinical diagnosis. We systematically evaluate several state-of-the-art VLMs, including general-purpose and medical-specific models, and reveal a substantial performance gap: the best-performing model achieves only 55% accuracy, far below clinical requirements. Our analysis identifies critical limitations of current VLMs in fetal ultrasound interpretation, highlighting the urgent need for domain-adapted architectures and specialized training approaches. Fetal-Gauge establishes a rigorous foundation for advancing multimodal deep learning in prenatal care and provides a pathway toward addressing global healthcare accessibility challenges. Our benchmark is publicly available *here*.

## 1 Introduction

Ultrasound is the primary modality for monitoring fetal health. In 2024, more than 132 million babies were born worldwide (Database Earth (2025)), and most pregnancies involved multiple ultrasound scans, typically averaging about 6–7 scans over the course of routine antenatal visits, with some women receiving as many as 8–10 scans (Susu et al. (2025)), depending on maternal health and resource availability. Ultrasound can detect up to 85% of major fetal anomalies (Dulgheroff et al. (2019)), highlighting its critical role in prenatal care. However, the growing reliance on ultrasound has intensified the global shortage of trained sonographers (Won et al. (2024)). Expanding the workforce alone may not meet demand, given the time and resources required for training. This challenge underscores the need for innovative solutions to ensure universal access to high-quality prenatal imaging.

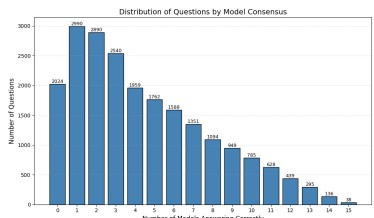

Figure 1: Number of questions categorized by the number of models answering correctly. Each bar represents how many of the 15 models answered a question correctly. For example, the bar at 15 represents the set of questions that all models answered correctly, comprising only 38 correctly answered questions out of a total of 21,468.

---

[*]Email: Hussain.Alasmawi@mbzuai.ac.ae

Deep learning (DL) methods have shown great promise in ultrasound interpretation, with advances in tasks such as plane classification (Maani et al. (2025)), biometric measurement (Qazi et al. (2023)), and anomaly detection (Arnaout et al. (2021), Taratynova et al. (2025)). DL has the potential to enhance efficiency by reducing scan time, improving image quality, and supporting the training of new sonographers. Among emerging DL approaches, Vision–Language Models (VLMs) stand out for their ability to jointly process visual and textual information. These models enable a wide range of tasks, including image classification, segmentation, report generation, and interactive question answering. While several benchmarks exist to evaluate VLMs in medical imaging, there is no benchmark dedicated to the fetal ultrasound domain, primarily due to its challenging nature, dependence on operator expertise, and the limited public availability of annotated datasets. To address this gap, we present the first comprehensive benchmark for evaluating VLMs in fetal ultrasound interpretation. We introduce the first large-scale benchmark for fetal ultrasound VLM evaluation, constructed by integrating 13 publicly available datasets. Our contributions are:

- **Fetal-Gauge dataset:** A benchmark of 42,036 images and 93,451 question–answer pairs, spanning diverse anatomical regions, clinical tasks, and question types. This is the first and largest dataset enabling reproducible evaluation of VLMs in fetal ultrasound.

- **Comprehensive VLM evaluation:** We systematically benchmark *15* state-of-the-art VLMs, including general-purpose (open and closed-source) and medical-specific, under a unified evaluation framework highlighting their capabilities and limitations in assessing fetal ultrasound.

- **Critical performance analysis**: We analyze performance for fetal ultrasound interpretation, including visual grounding of different structure sizes, handling of phantom images, and qualitative error patterns, providing key insights into their limitations.

This benchmark establishes a foundation for multimodal medical DL in fetal ultrasound, highlighting open challenges and setting the stage for future progress in applying VLMs to real-world clinical tasks.

## 2 RELATED WORK

### 2.1 MEDICAL VISUAL QUESTION ANSWERING DATASETS

Medical Visual Question Answering (Med-VQA) has grown rapidly in recent years, producing datasets to address challenges in clinical image interpretation. Early efforts such as VQA-Med (Ben Abacha et al. (2019)) (4,200 radiology images, 15,292 QAs) and VQA-RAD (Lau et al. (2018)) (3,515 QAs from 315 radiology cases) established structured question categories and manual curation by clinical experts. SLAKE (Liu et al. (2021)) expanded to a bilingual knowledge-based setting, with 14,028 Q&As over 642 radiology images annotated with rich semantic labels.

To improve scalability, PMC-VQA (Zhang et al. (2023b)) leveraged figure–caption pairs from medical publications to generate 227k VQA pairs from 149k images, though its reliance on paper figures introduces noise and limited clinical realism. More recent large-scale datasets, such as OmniMed-VQA (Hu et al. (2024)) (12 modalities, all from real clinical scenarios) and CAREs (Xia et al. (2024)) (16 modalities with evaluation of confidence, fairness, and safety), reflect a shift toward multimodal, multi-metric evaluation. Specialized datasets like PathVQA (He et al. (2020)) demonstrate the benefits of domain-specific collection.

However, no public fetal ultrasound VLM dataset exists, despite ultrasound being the primary imaging modality in prenatal care worldwide. Existing medical datasets concentrate on adult imaging modalities (CT, MRI, X-ray, histopathology), overlooking a domain that demands reasoning over noisy, operator-dependent, and contains various anatomical views. This omission creates a critical bottleneck for standardized benchmarking and hinders the development of multimodal medical DL systems capable of addressing clinically important tasks in fetal health, such as anomaly detection, gestational age estimation, and automated reporting in prenatal care.

## 2.2 VISION-LANGUAGE MODELS IN MEDICAL IMAGING

Recent advances in Vision-Language Models have catalyzed significant progress in medical image understanding, with specialized medical VLMs emerging to address domain-specific challenges. These models generally follow two primary approaches: curriculum learning with medical data adaptation and retrieval-augmented architectures.

Within the first approach, several models employ staged training strategies to adapt general vision-language capabilities to medical domains. LLaVA-Med (Li et al. (2023)) pioneered cost-efficient biomedical adaptation by combining PubMed figure-caption datasets with GPT-4 self-instruction, using a two-stage curriculum that first aligns biomedical vocabulary and then masters conversational semantics. However, its reliance on publication-derived data introduces quality limitations due to inherent noise and compression artifacts. MedVLM-R1 (Pan et al. (2025)) advances this approach by incorporating explicit reasoning generation, achieving remarkable improvements from 55.11% to 78.22% accuracy across MRI, CT, and X-ray tasks using Group Relative Policy Optimization (Shao et al. (2024)) with only 600 training samples and 2B parameters.

Large-scale foundation models aim for broader modality coverage. Lingshu (Xu et al. (2025)) addresses medical VLM limitations through training across eight imaging modalities, including adult ultrasound, enabling cross-modal understanding and generalization. HuatuoGPT-Vision (Zhang et al. (2023a)) scales this approach further with a 34B parameter model trained on refined PubMed image-text pairs across multiple modalities, representing one of the largest medical vision-language models available.

Specialized architectures also emerge for domain-specific optimization. MedGemma (Sellergren et al. (2025)) combines retrieval techniques with fine-tuned Gemma 2 models, providing broad specialty coverage across radiology, dermatology, pathology, and ophthalmology while emphasizing research accessibility.

Despite these advances, no medical VLM has been systematically evaluated on fetal ultrasound. The modality poses unique technical challenges for VLMs: integrating fine-grained spatial reasoning, interpreting images with substantial inter-operator variability, and coping with artifacts absent in other medical imaging modalities, such as standardized imaging. Without structured, targeted benchmarks, these limitations remain invisible, hindering progress toward clinically useful multimodal DL in prenatal care.

## 3 THE FETAL-GAUGE BENCHMARK

To address the critical gap in standardized evaluation for vision-language models in fetal ultrasound, we introduce Fetal-Gauge, a large-scale, multi-task VLM benchmark. This section details its construction. Section 3.1 defines the five core clinical tasks Fetal-Gauge is designed to evaluate. Section 3.2 outlines the data curation and standardization pipeline. Section 3.3 presents the data splitting strategy and final dataset statistics. Finally, section 3.4 discusses the importance of phantom images in our dataset.

### 3.1 TASK DESIGN

Fetal-Gauge is structured around five clinically distinct tasks and designed to assess a model's capabilities, from high-level scene understanding to fine-grained anatomical localization. Each task is formulated as a multiple-choice question (MCQ), a format chosen for its simplicity, amenability to straightforward evaluation, and ability to reduce the ambiguity inherent in free-text responses, thereby ensuring objective, scalable, and automated assessment. This structured approach also minimizes biases associated with open-ended responses, prevents hallucinations, and improves the fairness of model assessment. Figure 2 shows a sample of the images of our dataset, including the question we are using for each category. The tasks are:

- **Anatomical Fetal Plane Identification (PI):** Models must identify the specific anatomical plane shown in the ultrasound image (e.g., abdominal, trans-thalamic). This task evaluates fundamental image recognition and classification capabilities. In addition, this is an essential clinical task performed by a sonographer during the assessment of fetal growth.

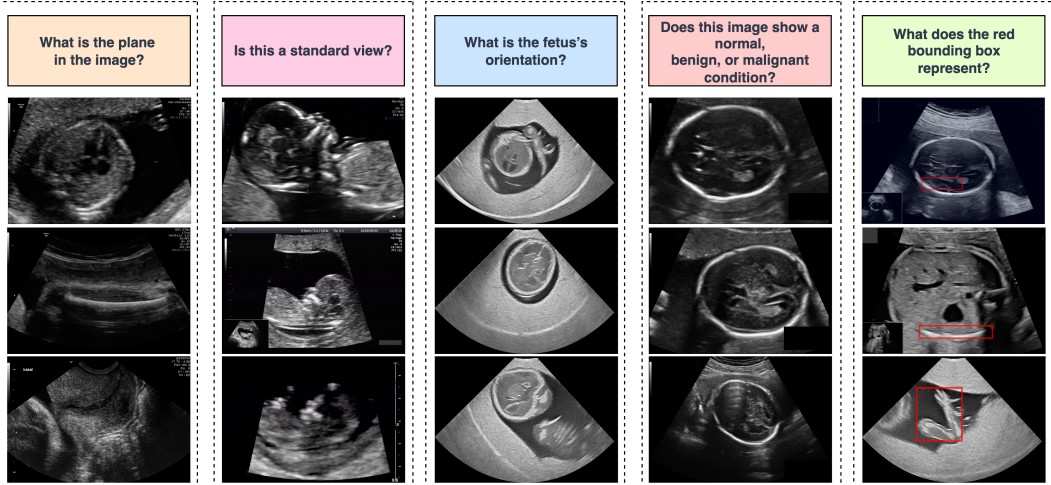

Figure 2: Question types in the dataset, along with representative sample images for each type, highlighting the diversity of visual content and associated questions.

- **Clinical View Conformity (VC):** Models must determine if an image meets the criteria of a clinically accepted standard view, reflecting their ability to assess adequacy for diagnostic use. This is an essential clinical task performed by sonographers to confirm that images are diagnostically adequate, ensuring accurate biometric measurements, reliable anomaly detection, and standardized reporting.

- **Fetal Orientation Assessment (FO):** This task requires the model to determine the orientation of the fetus within the scan, a crucial step in clinical assessment. Clinically, knowing fetal orientation is crucial for establishing presentation, guiding measurement techniques, detecting positional abnormalities, and aiding delivery planning and parental counseling.

- **Clinical Diagnosis (CD):** This task involves classifying the image as showing a normal, benign, or malignant condition, evaluating the model's capacity for clinical diagnostic reasoning. In practice, accurate classification enables timely clinical decision-making, appropriate referrals, and patient counseling, supporting management strategies and ensuring that abnormalities are promptly identified and addressed.

- **Visual Grounding of Anatomical Structures (VG):** Given an image with a red bounding box, the model must identify the anatomical structure highlighted within it. This task directly assesses the model's spatial reasoning and fine-grained object recognition. Clinically, precise localization of structures is fundamental for measurement, monitoring fetal development, guiding image-based interventions, and improving inter-observer consistency in assessments.

## 3.2 DATASET CURATION AND STANDARDIZATION

The construction of Fetal-Gauge followed a systematic pipeline to unify disparate data sources into a cohesive benchmark.

**Source Aggregation.** We began by aggregating thirteen publicly available fetal ultrasound datasets (detailed in the Appendix Table 4). This multi-source approach was crucial for ensuring diversity in imaging conditions, ultrasound machinery, hospital protocols, and patient demographics, thereby promoting the development of generalizable models.

**Task and Annotation Unification** . One of the main challenges was standardizing heterogeneous annotation types (e.g., image-level labels, segmentation masks, and bounding box annotations) into a unified format. For datasets containing segmentation masks, each mask was converted into a visual grounding task by extracting its bounding box coordinates. The bounding box was then overlaid onto the image as a red rectangle, enabling the formulation of spatially-grounded questions (e.g., "What does the red bounding box represent?").

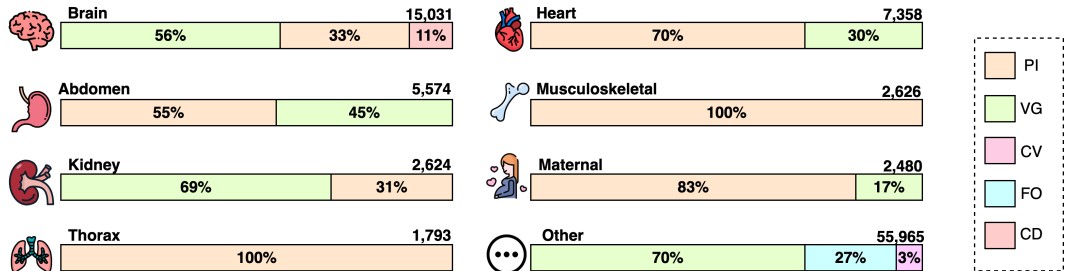

Figure 3: Distribution of benchmark tasks across anatomical regions. Colored segments represent question categories (legend on the right), with proportions shown within each bar and total counts indicated on the right. PI: Anatomical Plane Identification, VG: Visual Grounding of Anatomical Structures, VC: Clinical View Conformity, FO: Fetal Orientation Assessment, CD: Clinical Diagnosis.

**Vocabulary Normalization.** To minimize label noise from heterogeneous sources, we standardized the answer vocabulary. This involved expanding clinical abbreviations (e.g., "abdomcirc" to "abdominal circumference plane") and harmonizing synonymous terms to ensure consistency across the entire benchmark. For certain datasets, the specific imaging plane within an organ was not indicated—for example, the heart was labeled simply as "heart plane" rather than specifying views such as "Three-vessel plane" or "Four-chamber plane," and similar omissions occurred for the brain and abdomen. In such cases, we retained a generic "[organ] plane" label to ensure consistent terminology.

### 3.3 DATA PARTITIONING AND STATISTICS

**Splitting Strategy.** To ensure that model evaluation reflects true generalization capabilities rather than patient-specific memorization, we adopted a rigorous splitting strategy. Where available, we preserved the original train-test splits from the source datasets. For datasets without a prespecified split, we enforced strict patient-wise splits to prevent data leakage. Furthermore, to robustly assess generalization to unseen data distributions, datasets with limited sample sizes were allocated exclusively to the test set (detailed in Appendix Table 3). During this process, we curated the dataset by excluding classes with limited clinical value (e.g., "other"), focusing the benchmark on well-defined and meaningful clinical tasks.

**Dataset Scale.** The Fetal-Gauge benchmark is the most extensive collection of fetal ultrasound VLM data to date, comprising 42,036 images and 93,451 question-answer pairs. This includes a significant portion of phantom images (19k) for specialized task evaluation. The scale and diversity of Fetal-Gauge provide a robust foundation for training and comprehensively evaluating modern vision-language models. Figure 3 provides the distribution of our dataset per anatomy.

### 3.4 THE ROLE OF PHANTOM DATA

A substantial portion of Fetal-Gauge (19k images) is composed of data from anatomical phantoms. This is not a limitation but a strategic feature of our benchmark. Phantoms are the standard-of-care for training sonographers, allowing them to develop probe handling skills and learn to recognize standard planes in a controlled, repeatable environment. By including this data, we enable the development of DL systems designed for clinical practice, education, and simulation. This creates a pathway for future work where DL models could be trained and validated on phantoms before being deployed, or even serve as interactive training aids for human novices.

## 4 EVALUATIONS AND RESULTS

This section presents a comprehensive evaluation of state-of-the-art VLMs on the Fetal-Gauge benchmark. We first detail the experimental setup and then provide a multi-faceted analysis of model performance, both overall and on a per-task basis.

## 4.1 EVALUATION SETUP

**Evaluated Models.** We selected 15 prominent VLMs for evaluation. This cohort includes six models with a specialization in medical imaging (Lingshu-7B (Xu et al. (2025)), Lingshu-32B, MedVLM-R1 (Pan et al. (2025)), MedGemma-4b-it (Sellergren et al. (2025)), MedGemma-27b-it, HuatuoGPT-Vision-7B (Zhang et al. (2023a))), eight leading general-purpose models (InternVL3-8B-Instruct (Zhu et al. (2025)), InternVL3-14B-Instruct, Llama-3.2-11B-Vision-Instruct (Dubey et al. (2024)), Qwen2.5-VL-7B-Instruct (Wang et al. (2024)), Qwen2.5-VL-32B-Instruct, Aya-Vision-8b (Dash et al. (2025)), Aya-Vision-32b, vip-llava-7b (Cai et al. (2024))) and one commercial model GPT-5 (OpenAI (2025)). A random guess baseline was added to assess whether the model's high performance reflects real understanding rather than random chance. Additionally, we fine-tuned Qwen2.5-VL-7B-Instruct and Llama-3.2-11B-Vision-Instruct on our training set using LoRA (Hu et al. (2021)) for various amounts of epochs (referred to as model_x where x represents the number of epochs) to evaluate the impact of domain-specific training on model performance.

**Evaluation Protocol.** Given the multiple-choice question (MCQ) format of Fetal-Gauge, we use accuracy as the primary evaluation metric. Performance is measured across the entire test set on a per-task basis to enable a granular analysis.

## 4.2 OVERALL PERFORMANCE ANALYSIS

Our comprehensive evaluation reveals that fetal ultrasound interpretation poses significant challenges for current VLMs. As illustrated in Figure 1, the distribution of correct answers across our 15 evaluated models shows concerning patterns: very few questions were answered correctly by all models, with the majority of questions being answered correctly by only a small subset of models. This suggests fundamental limitations in current VLM architectures for this domain.

Table 1 presents the detailed performance breakdown across all five evaluation tasks. The results demonstrate a clear performance hierarchy: GPT-5 achieves the highest overall accuracy at 55%, followed by the Lingshu models (32B: 46%, 7B: 40%), while most other models perform at or near random chance levels (26% overall).

Table 1: Model accuracy across question types. Best accuracy per column is in **bold**, second-best is underlined. The second block of rows corresponds to fine-tuned models.

| Model | PI | VC | FO | CD | VG | Overall |
|---|---|---|---|---|---|---|
| RANDOM GUESS | 0.26 | 0.47 | 0.24 | 0.35 | 0.25 | 0.26 |
| AYA-VISION-32B | 0.19 | 0.56 | 0.22 | 0.39 | 0.17 | 0.19 |
| AYA-VISION-8B | 0.20 | 0.56 | 0.23 | 0.20 | 0.19 | 0.20 |
| HUATUOGPT-VISION-7B | 0.33 | 0.54 | 0.24 | **0.49** | 0.26 | 0.29 |
| INTERNVL3-14B-INSTRUCT | 0.25 | 0.51 | 0.25 | 0.39 | 0.17 | 0.21 |
| INTERNVL3-8B-INSTRUCT | 0.28 | 0.56 | **0.28** | 0.47 | 0.16 | 0.23 |
| LINGSHU-32B | 0.53 | 0.57 | 0.24 | 0.23 | 0.47 | 0.46 |
| LINGSHU-7B | 0.39 | 0.61 | 0.24 | 0.24 | 0.45 | 0.40 |
| LLAMA-3.2-11B-VISION-INSTRUCT | 0.40 | 0.55 | 0.23 | 0.23 | 0.31 | 0.33 |
| MEDGEMMA-27B-IT | 0.28 | 0.45 | 0.23 | 0.30 | 0.37 | 0.32 |
| MEDGEMMA-4B-IT | 0.32 | 0.44 | 0.22 | 0.22 | 0.27 | 0.28 |
| MEDVLM-R1 | 0.21 | 0.54 | 0.25 | 0.26 | 0.18 | 0.21 |
| QWEN2.5-VL-32B-INSTRUCT | 0.33 | 0.56 | 0.22 | 0.32 | 0.27 | 0.29 |
| QWEN2.5-VL-7B-INSTRUCT | 0.24 | 0.58 | 0.24 | 0.39 | 0.23 | 0.24 |
| VIP-LLAVA-7B | 0.29 | 0.46 | 0.25 | 0.36 | 0.23 | 0.26 |
| GPT-5 | **0.66** | **0.62** | 0.23 | 0.20 | **0.58** | **0.55** |
| LLAMA-3.2-11B-VISION-INSTRUCT_3 | 0.88 | 0.65 | 0.79 | 0.44 | 0.79 | 0.81 |
| LLAMA-3.2-11B-VISION-INSTRUCT_5 | **0.89** | 0.66 | 0.79 | 0.49 | 0.84 | 0.84 |
| LLAMA-3.2-11B-VISION-INSTRUCT_7 | **0.89** | 0.67 | 0.82 | 0.48 | **0.85** | 0.85 |
| LLAMA-3.2-11B-VISION-INSTRUCT_10 | **0.89** | 0.66 | **0.83** | 0.35 | **0.85** | 0.85 |
| QWEN2.5-VL-7B-INSTRUCT_3 | 0.45 | 0.70 | 0.45 | 0.49 | 0.52 | 0.49 |
| QWEN2.5-VL-7B-INSTRUCT_5 | 0.57 | 0.73 | 0.46 | **0.59** | 0.49 | 0.52 |
| QWEN2.5-VL-7B-INSTRUCT_7 | 0.37 | 0.71 | 0.43 | 0.56 | 0.43 | 0.42 |
| QWEN2.5-VL-7B-INSTRUCT_10 | 0.41 | **0.74** | 0.44 | 0.55 | 0.43 | 0.43 |

## 4.3 TASK-SPECIFIC PERFORMANCE

**Anatomical Plane Identification (PI):** This fundamental classification task reveals the largest performance variations among models. While most models struggle near random chance (26%), several demonstrate meaningful capabilities: GPT-5 leads with 66% accuracy, followed by Lingshu-32B (53%) and Llama-3.2-11B (40%).

**Clinical View Conformity (VC) & Fetal Orientation Assessment (CD):** All models performed at near-random levels on these two tasks, with none showing any meaningful performance.

**Visual Grounding of Anatomical Structures (VG):** Spatial localization tasks reveal clear performance tiers. GPT-5 achieves the highest accuracy (58%), followed by Lingshu models (32B: 47%, 7B: 45%). Most other models cluster near the 25% random baseline, suggesting fundamental limitations in fine-grained spatial reasoning.

## 4.4 DOMAIN ADAPTATION THROUGH FINE-TUNING

After task-specific fine-tuning, Llama-3.2-11B improves substantially from 33% to 85% overall accuracy, with consistent gains observed across all tasks. Qwen2.5-VL also shows clear improvements, increasing from 24% to 52% overall accuracy across the same set of tasks.

## 4.5 PHANTOM VS. CLINICAL PERFORMANCE

We evaluated model performance on phantom and real ultrasound images for two tasks: Anatomical Plane Identification (PI) and Visual Grounding of Anatomical Structures (VG). The evaluation dataset included 4,045 phantom VG questions, 1,146 phantom PI questions, 7,286 real VG questions, and 5,417 real PI questions.

Figure 4 presents the comparative results across models. Performance on phantom tasks was generally poor, with most models achieving accuracies close to or below the random guess baseline. In contrast, performance on real clinical images was substantially higher, with nearly all models surpassing random chance levels.

Among the models tested, Lingshu-32B and GPT-5 demonstrated the strongest performance. Lingshu-32B exceeded 50% accuracy on both PI and VG real-image tasks, while GPT-5 consistently ranked highest overall, with balanced performance across phantom and real domains.

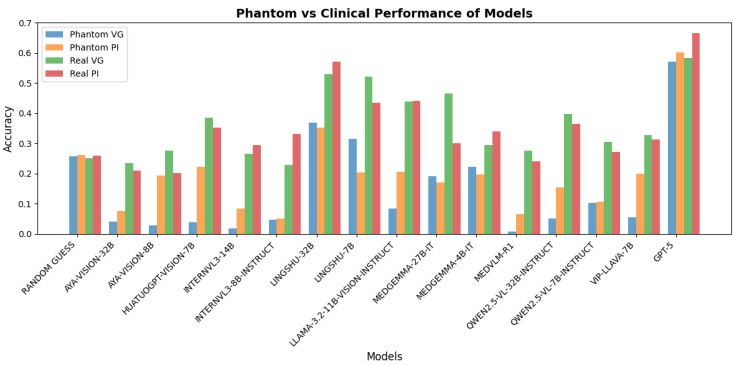

Figure 4: Bar plot presenting model accuracy on phantom and clinical ultrasound images through two questions ('Plane of the image' and 'Bounding box label')

## 4.6 IMPACT OF ANATOMICAL STRUCTURE SIZE

To further evaluate visual grounding, we analyzed performance based on bounding box size. Our evaluation set is composed of 7,373 small, 1,799 medium, and 2,160 large questions. Results are summarized in Table 2. Models performed best on large structures, with accuracies often exceeding 80%. However, performance dropped sharply for medium and small targets, where accuracies were frequently below 50%.

## 5 ANALYSIS

**Commercial advantage of GPT-5.** GPT-5 consistently outperformed all other models across tasks, achieving the highest accuracy in both phantom and clinical datasets. As a closed-source commercial model trained on large-scale proprietary data, it is possible that its training distribution included fetal ultrasound images or closely related medical data. This may explain its superior performance compared to open-source models, which lack access to such data.

Table 2: Performance comparison of models based on bounding box size. The best results are in **bold** and the second-best are underlined.

| Model | Small | Medium | Large |
|---|---|---|---|
| AYA-VISION-32B | 0.22 | 0.09 | 0.04 |
| AYA-VISION-8B | 0.20 | 0.11 | 0.20 |
| HUATUOGPT-VISION-7B | 0.24 | 0.14 | 0.45 |
| INTERNVL3-14B | 0.20 | 0.08 | 0.20 |
| INTERNVL3-8B-INSTRUCT | 0.19 | 0.13 | 0.10 |
| LINGSHU-32B | 0.38 | 0.45 | 0.79 |
| LINGSHU-7B | 0.34 | 0.45 | 0.82 |
| LLAMA-3.2-11B-VISION-INSTRUCT | 0.29 | 0.18 | 0.51 |
| MEDGEMMA-27B-IT | 0.29 | 0.32 | 0.67 |
| MEDGEMMA-4B-IT | 0.21 | 0.31 | 0.43 |
| MEDVLM-R1 | 0.25 | 0.04 | 0.06 |
| QWEN2.5-VL-32B-INSTRUCT | 0.29 | 0.13 | 0.35 |
| QWEN2.5-VL-7B-INSTRUCT | 0.23 | 0.19 | 0.28 |
| VIP-LLAVA-7B | 0.23 | 0.13 | 0.30 |
| GPT-5 | **0.48** | **0.67** | **0.85** |

**Limited utility of existing medical VLMs.** None of the evaluated medical VLMs reported training on fetal ultrasound data, which likely explains their limited performance. While they leverage adult MRI and CT scans, these modalities differ significantly in appearance and resolution from fetal ultrasound. Nevertheless, such data still provide anatomical priors closer to fetal imaging than natural image distributions (e.g., cars, trees, animals). This partial domain relevance likely contributed to the modest but still insufficient performance observed in these models.

**Role of ultrasound-specific training.** The Lingshu models represent the only group that explicitly reported training on adult ultrasound data. This appears to have been critical for their relatively strong performance among open-source models. In particular, Lingshu-32B achieved over 50% accuracy in Anatomical Plane Identification and Visual Grounding on real images, suggesting that exposure to ultrasound imaging, even of adults, provides transferable knowledge that aids generalization to fetal ultrasound.

**Domain adaptation and fine-grained localization.** The phantom vs. clinical comparison underscored the persistent domain adaptation gap, with phantom images proving particularly challenging for all models. Additionally, bounding box analysis revealed that models are far more successful at grounding large anatomical structures than small or medium ones. This highlights an ongoing weakness in fine-grained localization, which is critical for clinical tasks requiring precise anatomical identification.

## 5.1 QUALITATIVE ANALYSIS

To better understand the limitations of the models, we performed a qualitative analysis of challenging cases where many models provided incorrect answers. Figure 5 showcases several of these instances from both clinical and phantom datasets.

**Case A: Clinical Image Challenges** In the first case (Figure 5A, left), most models failed to identify the four-chamber plane correctly. The view is significantly zoomed out, making the key anatomical features less distinct. We observe that many models incorrectly selected the transventricular plane and the transverse kidney plane, suggesting that they were confused by the general elliptical shape present in all three planes, rather than identifying the specific internal structures.

Similarly, in the second image (Figure 5A, center), no model correctly identified the transverse kidney plane, despite its straightforward presentation. Instead, models predominantly chose the transcerebellar plane and four-chamber plane, which also present as elliptical shapes. This indicates a potential model bias towards more commonly encountered elliptical structures.

The third example (Figure 5A, right) shows that while about half of the models correctly identified the transcerebellar plane, a significant number chose the transthalamic plane. This confusion is understandable, as these two planes are anatomically close and can be challenging to differentiate, even for a novice sonographer.

**Case B: Phantom Image Challenges** The images in Case B were sourced from a phantom. In the first image (Figure 5B, left), models were asked to identify the structure within the red bounding box, which is the abdomen. However, many models were incorrect, selecting options like cerebellum,

midbrain, and head. This suggests that the models were heavily influenced by the global context of the image (which includes prominent brain structures) and did not focus exclusively on the specified region of interest (ROI).

The final two examples (Figure 5B, center and right) further demonstrate the models' difficulties. Many failed to identify the legs and the femur plane correctly. This poor performance can be attributed to several factors: the structures of interest are small, the images are zoomed out, and the phantom images are significantly brighter than typical clinical ultrasounds. This brightness variation likely represents an out-of-distribution characteristic that the models were not adequately trained to handle.

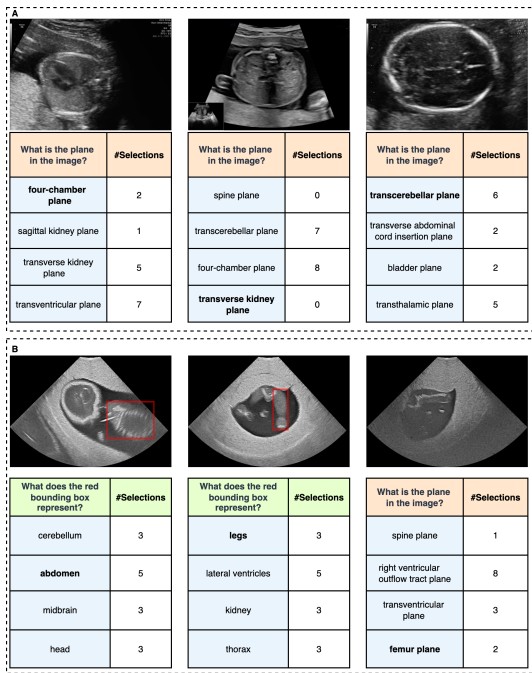

Figure 5: Examples of challenging cases illustrating model failure modes. The figure shows model predictions for (A) three clinical and (B) three phantom ultrasound images. For each case, a table displays the distribution of answers from 15 models, with the correct option shown in bold. These examples highlight common errors, such as confusion between similarly shaped structures and poor generalization to phantom images.

# 6 CONCLUSION

We introduce Fetal-Gauge, the first large-scale benchmark for evaluating Vision-Language Models in fetal ultrasound interpretation, comprising 42,036 images and 93,451 question-answer pairs across five clinical tasks. Our systematic evaluation of 15 state-of-the-art VLMs reveals substantial limitations: the best-performing model achieves only 55% accuracy, far below clinical requirements.

Key findings highlight critical gaps in current VLM capabilities. Models struggle with fine-grained spatial reasoning, particularly for small anatomical structures, and show poor domain adaptation between phantom and clinical images. While ultrasound-specific training data (as in Lingshu models) improves performance, fundamental architectural limitations persist across all evaluated models.

Our benchmark establishes a foundation for developing specialized VLMs in prenatal care and reveals urgent research priorities: ultrasound-specific architectures, improved spatial reasoning, and robust domain adaptation strategies. The substantial performance gaps underscore both current limitations and opportunities for methodological innovation in medical multimodal DL. Fetal-Gauge provides the rigorous evaluation framework necessary for measuring progress toward clinically viable fetal ultrasound interpretation systems.

THE USE OF LARGE LANGUAGE MODELS

During the preparation of this work, the authors used ChatGPT to enhance writing. After using this tool/service, the authors reviewed and edited the content as needed and take full responsibility for the content of the publication.

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

# 7 APPENDIX

Table 3: Dataset-wise distribution of images and corresponding questions across training and testing splits. The values represent the number of images, with the number of associated questions shown in parentheses.

| Dataset Name | Train | Test | Total |
|---|---|---|---|
| 3-Vessel View Dataset Belciug (2022) | 0(0) | 253(253) | 253(253) |
| FASS Da Correggio et al. (2023) | 1,265(4,983) | 323(1,276) | 1,588(6,259) |
| Echo (1st Trimester) Stoean et al. (2021) | 3,223(3,223) | 690(690) | 3,913(3,913) |
| Echo (2nd Trimester) Stoean et al. (2022) | 281(756) | 94(250) | 375(1,006) |
| Fetal Planes Burgos-Artizzu et al. (2020) | 2,908(2,908) | 2,187(2,187) | 5,095(5,095) |
| Fetal Planes & Organs Belciug (2024) | 2,776(10,447) | 695(2,704) | 3,471(13,151) |
| Fetus Head Tumor Anitha (2024) | 1,420(1,420) | 249(249) | 1,669(1,669) |
| FOCUS Wu et al. (2025) | 250(750) | 50(150) | 300(900) |
| FPUS23 Prabakaran et al. (2023) | 15,248(31,433) | 3,812(7,857) | 19,060(39,290) |
| Large Fetal Head Biometry Alzubaidi et al. (2023) | 2,332(5,665) | 1,715(4,337) | 4,047(10,002) |
| MFUP Sendra-Balcells et al. (2023) | 217(217) | 233(233) | 450(450) |
| NatalIA González et al. (2024) | 0(0) | 346(346) | 346(346) |
| NT Scan Cui & Dong (2022) | 1,372(10,181) | 312(936) | 1,684(11,117) |
| **Total** | **31,292(71,983)** | **10,959(21,468)** | **42,036(93,451)** |

Table 4: Summary of fetal ultrasound datasets with descriptions, annotation types, and annotated anatomical structures or planes.

| Dataset | Description | Annotation | Annotated Structures / Labels |
|---|---|---|---|
| 3-Vessel View Dataset | Ultrasound images of the 3 vessels & gallbladder of second trimester fetuses. | Classification | 3-vessel & bladder |
| FASS | 2D ultrasound of fetal abdomen (term pregnancies) with manual segmentation of abdominal organs (aorta, umbilical vein, stomach, liver) to support prenatal diagnostics. | Segmentation | Abdominal aorta, intrahepatic umbilical vein, stomach, liver area |
| Echo (1st Trimester) | Frames from fetal cardiac sweep videos (12–14 weeks GA, Doppler color) labeled by view. Contains 6,720 images across four main cardiac planes. | Classification | Atrioventricular flow (4-chamber), Aorta (LVOT), Great vessels (RVOT), Arterial arches (3-vessel), plus "other" |
| Echo (2nd Trimester) | Dataset of 8 ultrasound video sweeps from fetuses (21–24 weeks GA) yielding 1,040 frames across four standard heart views. Each frame is segmented with 11 cardiac structures. | Segmentation | 11 structures: e.g., septum (S), right atrium (RA), left atrium (LA), aorta (Ao), right ventricle (RV), etc. |
| Fetal Planes | Large screening dataset (mid-second Trimester) from two hospitals. 12,400+ images from 1,792 patients, labeled into six classes: four fetal planes (Abdomen, Brain, Femur, Thorax), Cervix, and Other. (Brain images are further sub-classified into three views for fine-grained analysis.) | Classification | Abdomen, Brain (trans-thalamic, trans-cerebellum, trans-ventricular), Femur, Thorax, Cervix, Other |
| Fetal Planes & Organs | 2D US scans of fetal morphology. They are divided into different view planes, and the organs are segmented. | Segmentation & Classification | 11 fetal ultrasound planes (e.g., biparietal head, abdominal, heart) and 18 annotated structures (e.g., bladder, aorta, kidney, cerebellum). |
| Fetus Head Tumor | Ultrasound of fetal head that contains normal, benign, and malignant cases. Each image is annotated at the frame level with one of three diagnostic. | Segmentation & Classification | Fetal head |
| FOCUS | 4-chamber fetal heart images (second Trimester) with manual segmentation of heart and thorax regions for biometric measurement (e.g., cardiothoracic ratio). | Segmentation | Cardiac chambers and thoracic regions |
| FPUS23 | Phantom fetal ultrasound at 23 weeks GA. 15,728 images for tasks: plane identification, fetus orientation, anatomical features, and bounding-box detection. | Classification (plane/orientation/features) and Detection | Diagnostic planes, fetal orientation, anatomical landmarks, bounding-boxes of anatomy |
| Large Fetal Head Biometry | High-res fetal head ultrasound images annotated by experts for brain biometry. Used for training segmentation/biometry algorithms. | Segmentation | Fetal brain, cavum septum pellucidum (CSP), lateral ventricles (LV) |
| MFUP | Screening images from 5 African centers (low-resource settings). Contains routine second-trimester scans labeled into four common fetal planes (Abdomen, Brain, Femur, Thorax). | Classification | Abdomen, Brain, Femur, Thorax |
| NatalIA | Phantom scans by non-experts at 23 weeks GA. 19,407 frames from 90 free-hand videos (POCUS device) simulating low-resource scans. Each frame is labeled for fetal plane (including "no-plane"). | Classification | Biparietal head plane, Abdominal plane, Heart plane, Spine plane, Femur plane, No-plane |
| NT Scan | Sagittal ultrasound images (11–14 weeks GA) for NT measurement plane classification and key structure detection (Down syndrome screening). | Classification and Object Detection | Thalami, midbrain, palate, 4th ventricle, cisterna magna, nuchal translucency, nasal tip, nasal skin, nasal bone |

Table 5: Summary of datasets with gestational age, ultrasound machines, license, patient counts, and geographic distribution.

| Dataset Name | Gestational Age | Machine | License | #Patients | Geographic Distribution |
|---|---|---|---|---|---|
| 3-Vessel View Dataset | 2nd Trimester | Not Provided | CC BY 4.0 | 15 | Romania – University Emergency County Hospital of Craiova |
| FASS | Not Provided | Siemens Acuson; GE Voluson 730; Philips EPIQ Elite | CC BY 4.0 | 169 | Brazil – University Hospital Polydoro Ernani de São Thiago, Florianópolis |
| Echo (1st Trimester) | 1st Trimester | GE Voluson E10; GE Voluson E8; GE Voluson E6 | CC BY 4.0 | 326 | Not Provided |
| Echo (2nd Trimester) | 2nd Trimester | Not Provided | CC BY 4.0 | 8 | Not Provided |
| Fetal Planes | 2nd & 3rd Trimester | GE Voluson E6; GE Voluson S8; GE Voluson S10; Aloka | CC BY 4.0 | 1,792 | Spain – Hospital Clinic and Hospital Sant Joan de Déu, Barcelona |
| Fetal Planes & Organs | 2nd Trimester | GE LOGIQ e; GE Voluson 730 Pro | CC BY 4.0 | 215 | Romania – University Emergency County Hospital of Craiova |
| Fetus Head Tumor | Not Provided | Not Provided | CC BY 4.0 | Not Provided | Not Provided |
| FOCUS | 2nd Trimester | Not Provided | CC BY 4.0 | Not Provided | Not Provided |
| FPUS23 | 1st Trimester | Philips EPIQ 7 | Not Provided | N/A | N/A |
| Large Fetal Head Biometry | 2nd & 3rd Trimester | GE Voluson E8; GE Voluson 730 | CC BY 4.0 | 551 | Netherlands – Radboud University Medical Center, Nijmegen |
| MFUP | 2nd & 3rd Trimester | GE Medical Systems; Siemens; Edan Instruments; Mindray; Aloka | CC BY 4.0 | 125 | Egypt; Algeria; Uganda; Ghana; Malawi |
| NatalIA | 2nd Trimester | Clarius C3 HD3 (POCUS) | CC BY 4.0 | N/A | N/A |
| NT Scan | 1st Trimester | Not Provided | CC BY 4.0 | 1,519 | China – Shenzhen People's Hospital |

Table 6: Dataset composition across fetal ultrasound tasks.

| Task | Acronym | # Samples |
|---|---|---|
| Visual Grounding of Anatomical Structures | VG | 54,601 |
| Plane Identification | PI | 20,131 |
| Fetal Orientation Assessment | FO | 15,113 |
| Clinical View Conformity | VC | 1,684 |
| Clinical Diagnosis | CD | 1,669 |

