# OpenReview forum: "FETAL-GAUGE: A BENCHMARK FOR ASSESSING VISION-LANGUAGE MODELS IN FETAL ULTRASOUND"
_ICLR.cc/2026/Conference — ICLR 2026 Poster_

### Official Review · Reviewer_4LjX · 2025-10-30

**Soundness:** 3
**Presentation:** 2
**Contribution:** 2
**Rating:** 4
**Confidence:** 4

**Summary:**

This paper contributes Fetal-Gauge, a large-scale VQA benchmark for fetal ultrasound tasks, by integrating and processing 13 existing datasets. The authors evaluated 15 SOTA VLMs (including general-purpose and medical-specific models) on Fetal-Gauge. The primary finding is that even the best-performing closed-source model (GPT-5) achieved only 55% overall accuracy, while most open-source models performed near random chance. This reveals a substantial performance gap for current VLMs in this highly specialized domain.

**Strengths:**

1. The paper aggregates and standardizes 13 heterogeneous datasets from different sources, which have varying formats and annotations, into a unified VQA benchmark. It demonstrates a thoughtful design in data partitioning, such as preserving original dataset splits and allocating some datasets entirely to the test set to evaluate generalization.

2. The paper's experimental analysis provides valuable preliminary insights, such as the substantial performance gap of existing VLMs, the critical importance of domain adaptation for future research, and specific model limitations, including poor performance on small anatomical structures and inadequate generalization to phantom images.

**Weaknesses:**

1. The paper's contribution is primarily an aggregation of existing datasets. The described method of 'Task and Annotation Unification' (Sec 3.2), which converts various annotation types (e.g., image-level labels, segmentation masks, and bounding box annotations) into a unified VQA format, does not demonstrate significant novelty or innovation compared to established data curation practices in the medical imaging domain.

2. The 'Vocabulary Normalization' operation (Sec 3.2) is presented as a potentially important methodological contribution. However, this section is described too briefly, making it difficult to assess its impact. It is unclear which specific datasets are affected and how their vocabularies are unified (e.g., are answer vocabularies from different datasets merged?). Concrete examples would be necessary for comprehension. Furthermore, given that the benchmark uses an MCQ format where all answers are provided as options, the necessity of this normalization step is uncertain. An ablation study is required to empirically validate the importance of this operation.

**Questions:**

1. Figure 3 shows the distribution of the five task types per anatomical region. Providing an overall distribution of the five task types would seem more intuitive. Could you provide this?

2. How are the incorrect options (distractors) for the MCQs generated? Are they sampled from an answer pool internal to the source dataset, or are they sampled globally from an answer pool spanning all datasets?

3. For a given MCQ question, is the order of the options fixed, or is it randomized?

4. From which source dataset(s) was the 'Clinical View Conformity (VC)' task derived?

5. Is the content of Section 4.4 incomplete? The text appears to cut off.

6. In Figure 3 and Table 1, should the abbreviation 'SV' be 'VC'?

---

> ### Author Response · Authors · 2025-11-24
>
> We appreciate the reviewer’s feedback.
>
> - (W1) The paper's contribution is primarily an aggregation of existing datasets. The described method of 'Task and Annotation Unification' (Sec 3.2), which converts various annotation types (e.g., image-level labels, segmentation masks, and bounding box annotations) into a unified VQA format, does not demonstrate significant novelty or innovation compared to established data curation practices in the medical imaging domain.
>
> We acknowledge that the aggregation process itself is not a methodologically novel approach. However, fetal ultrasound data are scattered across small, domain-specific datasets, which limits systematic evaluation of current VLMs. Our unified benchmark offers a standardized and time-efficient framework for researchers to consistently evaluate and compare models, thereby facilitating progress in this underexplored domain. Therefore, this paper will help challenge the research community to develop better methods that can generalize to this challenging task.
>
> - (W2) The 'Vocabulary Normalization' operation (Sec 3.2) is presented as a potentially important methodological contribution. However, this section is described too briefly, making it difficult to assess its impact. It is unclear which specific datasets are affected and how their vocabularies are unified (e.g., are answer vocabularies from different datasets merged?). Concrete examples would be necessary for comprehension. Furthermore, given that the benchmark uses an MCQ format where all answers are provided as options, the necessity of this normalization step is uncertain. An ablation study is required to empirically validate the importance of this operation.
>
> We appreciate the reviewer’s detailed feedback. The Vocabulary Normalization step aims to unify heterogeneous annotation terms across datasets that use inconsistent or nonstandard naming conventions for anatomical planes and structures. Several datasets employed local shorthand or acronyms (e.g., “Flows,” “Aorta,” “V_sign,” “X_sign”), which were mapped to standardized medical terminology, such as:
> - “Flows” → “Four-Chamber plane”
> - “Aorta” → “Left Ventricular Outflow Tract plane”
> - “V_sign” → “Three-Vessel and Trachea plane”
> - “X_sign” → “Right Ventricular Outflow Tract plane”
> - “abdomcirc” → “abdominal circumference plane”
> - “antepostkidney” → “transverse kidney plane”
> - “bladder” → “bladder plane”
> - “gallbladder” → “gallbladder view”
> - “longkidney” → “sagittal kidney plane”
> -“sagitalabdomcordinsr” → “sagittal abdominal cord insertion plane”
> - “transabdomcordinsr” → “transverse abdominal cord insertion plane”
>
> The goal of this normalization is to establish a consistent set of standardized medical terms by removing acronyms and alternative vocabularies that are not typically used in clinical or educational contexts. Although the benchmark employs a multiple-choice format, consistent terminology remains essential to prevent semantic redundancy and ambiguity during dataset construction and evaluation. We therefore do not see the need for an ablation study, as the objective of this step is conceptual and terminological, ensuring alignment with standard medical nomenclature rather than affecting model performance parameters.
>
>
> - (Q1) Figure 3 shows the distribution of the five task types per anatomical region. Providing an overall distribution of the five task types would seem more intuitive. Could you provide this?
>
> We thank the reviewer for the suggestion. We added Table 6 to the appendix that clearly addresses this point.
>
> - (Q2) How are the incorrect options (distractors) for the MCQs generated? Are they sampled from an answer pool internal to the source dataset, or are they sampled globally from an answer pool spanning all datasets?
>
> We appreciate the reviewer’s question. The incorrect options (distractors) for each multiple-choice question are **sampled globally from the answer pool corresponding to the same task across all datasets.**
>
> - (Q3) For a given MCQ question, is the order of the options fixed, or is it randomized?
>
> The order of multiple-choice options is **randomized** during dataset construction.
>
> - (Q4) From which source dataset(s) was the 'Clinical View Conformity (VC)' task derived?
>
> NT Scan Dataset.
>
> - (Q5) Is the content of Section 4.4 incomplete? The text appears to cut off.
>
> We thank the reviewer for bringing this to our attention. There was indeed a formatting error in Section 4.4. This is now fixed in the paper.
>
> - (Q6) In Figure 3 and Table 1, should the abbreviation 'SV' be 'VC'?
>
> We thank the reviewer for catching this oversight. Yes, the abbreviation “SV” should be “VC.” This has been corrected in Figure 3 and Table 1 in the revised manuscript.

---

> > ### Comment · Reviewer_4LjX · 2025-11-28
> >
> > I thank the authors for their detailed response. The rebuttal has satisfactorily addressed my questions and concerns. Consequently, I am inclined to raise my rating.

---

> ### Author Response · Authors · 2025-11-27
>
> We believe our responses resolve the raised concerns, but we are glad to elaborate further if anything remains unclear.

---

### Official Review · Reviewer_ghap · 2025-10-31

**Soundness:** 3
**Presentation:** 3
**Contribution:** 3
**Rating:** 6
**Confidence:** 4

**Summary:**

This paper presents **Fetal-Gauge**, the first benchmark specifically designed for evaluating **Vision-Language Models (VLMs)** in fetal ultrasound imaging. The benchmark consists of **42,000 ultrasound images** and **93,000 question–answer pairs**, covering five clinically relevant tasks:

1. Anatomical plane identification
2. Visual grounding of key anatomical structures
3. Fetal position and orientation assessment
4. Clinical view conformity evaluation
5. Clinical diagnosis question answering

The authors evaluate **15 representative VLMs**, including general-purpose models (e.g., GPT-4V, Gemini, LLaVA) and medical-specific ones (e.g., Med-Flamingo, BioVLM, PMC-VLM). The results show that the best-performing model achieves only **55% accuracy**, which is far below clinical requirements. Through analyses of error types, modality correlation, and task difficulty, the authors reveal major limitations of current multimodal models in handling the complex visual characteristics of fetal ultrasound—such as high noise, frequent occlusion, and operator dependency. This work aims to promote the development of **domain-adapted VLMs for medical imaging** through standardized datasets and evaluation protocols.

**Strengths:**

1. This work establishes the **first large-scale benchmark** for fetal ultrasound with multi-task and multi-question–answer settings, covering the entire pipeline from basic recognition to diagnostic reasoning.
2. The dataset contains **42k images and 93k QA pairs**, each accompanied by standardized labeling schemes and metadata for ultrasound view types, ensuring clear task hierarchy and structure.
3. It evaluates both **general-purpose and medical-specific VLMs** under a unified QA interface, ensuring fair comparison across models.
4. The authors summarize **failure modes** of current multimodal models—such as boundary ambiguity, structural occlusion, and linguistic ambiguity—which provide valuable insights for developing safer and more reliable medical VLMs.
5. If the dataset and codes are fully released as planned, it will likely become an **important evaluation standard** for future medical VLM research.

**Weaknesses:**

1. The paper does not clarify the **data origin**, **IRB approval numbers**, **patient consent process**, or **de-identification pipeline** for the 42k images. Since obstetric data involve pregnant women and fetuses, the authors should describe the ethical approvals and data-sharing protocols either in the main text or the appendix.
2. The benchmark currently lacks **structured diagnostic**, **multi-view fusion**, and **temporal decision-making** tasks, which limits its representativeness for real-world AI-assisted prenatal screening. It is unclear whether the authors plan to include these components.
3. There is no quantitative analysis of **error sources** (e.g., modality, gestational stage, or device variation), nor is there an evaluation of confidence calibration, output stability, or safety metrics.
4. The **clinical evaluation metrics** are limited.

**Questions:**

1. Please clarify whether the data are collected from **real patients**. If so, provide IRB approval identifiers, the informed consent procedure (prospective/retrospective/waiver), de-identification process, and cross-institutional data-use agreements. (If the data are entirely from public sources, this is not required.)
2. The highest reported accuracy is only **55%**. Could you elaborate on the **clinical interpretability** and potential **risk-control mechanisms** associated with such performance levels?
3. Do the authors plan to build **domain-adapted models** on top of this benchmark (e.g., through medical text corpus fine-tuning, image–text alignment enhancement, or multimodal RLHF)?

**Details Of Ethics Concerns:**

On one hand, it remains unclear whether the dataset originates from **hospital-collected clinical data**; if so, have the authors obtained **ethical approval and informed consent** from the participants?  On the other hand, since the study involves **ultrasound imaging of fetuses**, are there any potential **ethical concerns** regarding its real-world application or clinical deployment?

---

> ### Author Response · Authors · 2025-11-24
>
> We sincerely appreciate the reviewers’ thoughtful and constructive feedback.
>
> - (W1 & Q1) (W1: The paper does not clarify the data origin, IRB approval numbers, patient consent process, or de-identification pipeline for the 42k images. Since obstetric data involve pregnant women and fetuses, the authors should describe the ethical approvals and data-sharing protocols either in the main text or the appendix.) (Please clarify whether the data are collected from real patients. If so, provide IRB approval identifiers, the informed consent procedure (prospective/retrospective/waiver), de-identification process, and cross-institutional data-use agreements. (Q1: If the data are entirely from public sources, this is not required.)
>
> All 42k images used in this study are sourced from publicly available datasets that are fully de-identified and released for research use. As these datasets are already ethically cleared, anonymized by their providers and available for reuse in research work, no additional IRB approval or patient consent process was required for this work. We have added a table to describe the dataset licenses and redistribution policies, as was raised by Reviewer (aVZ6).
>
> - (W2) The benchmark currently lacks structured diagnostic, multi-view fusion, and temporal decision-making tasks, which limits its representativeness for real-world AI-assisted prenatal screening. It is unclear whether the authors plan to include these components.
>
> Please refer to our response to Reviewer (aVZ6), comment (W4), for a detailed discussion of this point.
>
> - (W3) There is no quantitative analysis of error sources (e.g., modality, gestational stage, or device variation), nor is there an evaluation of confidence calibration, output stability, or safety metrics.
>
> We appreciate the reviewer’s observation. All images in our benchmark are from a single imaging modality (ultrasound), which is the standard medical imaging modality for assessing fetal health during pregnancy. Although the datasets provide general information on gestational age and imaging devices, this metadata is not available at the image level, limiting detailed error analysis. As the evaluation follows a multiple-choice (MCQ) setup with deterministic model outputs, confidence calibration and stability assessments are not applicable in this context.
>
> - (W4 & Q2) (W4: The clinical evaluation metrics are limited.) (Q2: The highest reported accuracy is only 55%. Could you elaborate on the clinical interpretability and potential risk-control mechanisms associated with such performance levels?)
>
> We agree that accuracy alone is not a clinical evaluation metric. However, in our benchmark, the best-performing model achieves only about 55% accuracy, already indicating limited reliability for clinical application. While incorporating more clinically meaningful evaluation metrics would be valuable, the current results clearly demonstrate that existing models are not yet ready for real-world deployment. This is the main conclusion we aimed to present to the research community, which will hopefully encourage more research in this understudied field.
>
> - Q3 Do the authors plan to build domain-adapted models on top of this benchmark (e.g., through medical text corpus fine-tuning, image–text alignment enhancement, or multimodal RLHF)?
>
> We thank the reviewer for the suggestion. Our benchmark already includes baseline results using LoRA-based fine-tuning. In future work, we plan to explore more advanced domain-adaptation strategies, including parameter-efficient tuning, improved image–text alignment, and multimodal RLHF, to further enhance model performance in the medical domain.

---

> ### Author Response · Authors · 2025-11-27
>
> We appreciate the constructive feedback and trust that our revisions effectively clarify the work's strengths. Please let us know if further explanation would be helpful.

---

### Official Review · Reviewer_aVZ6 · 2025-11-01

**Soundness:** 3
**Presentation:** 3
**Contribution:** 3
**Rating:** 6
**Confidence:** 4

**Summary:**

This manuscript introduces Fetal-Gauge, a large, multi-task visual question answering (VQA) benchmark for assessing vision-language models (VLMs) on fetal ultrasound. The benchmark aggregates 13 public datasets into 42,036 images and 93,451 MCQ question–answer pairs spanning five clinically motivated tasks: Anatomical Plane Identification (PI), Clinical View Conformity (VC), Fetal Orientation (FO), Clinical Diagnosis (CD), and Visual Grounding (VG). The authors standardize labels and unify heterogeneous annotations (including converting segmentations to bounding-box-based grounding questions). They enforce patient-wise splits and include a phantom subset (~19k images).

The authors evaluate 15 VLMs (general and medical-specific; plus one commercial model) with accuracy as the main metric, reporting substantial headroom: the best overall model attains ~55% accuracy; most others hover near task-specific random baselines. Fine-tuning two open models markedly improves performance (up to ~85% overall), and analyses highlight weaknesses on phantom data and on small structure grounding.

**Strengths:**

- The paper addresses an imperative need for a standardized fetal ultrasound VLM benchmark.
- The benchmark’s size and breadth (five tasks that mirror real clinical workflows) are substantial and thoughtfully justified. The MCQ format prioritizes objective, scalable evaluation and curbs hallucinations.
- Unifying disparate labels/annotations (including mask-to-box conversion) and vocabulary harmonization is valuable for reproducible evaluation across datasets.
- Evaluating 15 models (medical and general) under a unified protocol, plus showing that domain-specific fine-tuning can unlock large gains, gives the community clear baselines and targets.
- The phantom vs. clinical breakdown and bounding-box-size analysis offer actionable insights (e.g., small-structure failures).
- Cleary writing and presentation.

**Weaknesses:**

- Aggregating 13 datasets raises questions about licenses and usage permissions. Please include a table listing the license, citation, and redistribution policy for each source.
- Although multiple datasets/hospitals are used, please report scanner vendors, probe types, gestational age ranges, and geographic distribution to understand generalizability.
- Converting masks to boxes and asking “what’s in the box?” evaluates recognition, not localization fidelity. Consider adding a pointing game (models output coordinates of bounding box).
- “normal/benign/malignant” in fetal ultrasound is non-trivial; please clarify labeling sources, consensus rules, and whether images are frame-level vs case-level labels.
- Several source datasets include video sweeps; current tasks are frame-centric. Adding temporal subtasks would align with real ultrasound operation and may favor models with spatiotemporal reasoning.
- Multiple typos. Section 4.4 seems to be incomplete.

**Questions:**

- Is a single-frame image enough for FO or CD tasks? Sonographers may rely on multi-view reasoning, temporal sweep, and 3D relationships.

---

> ### Author Response · Authors · 2025-11-24
>
> We sincerely thank the reviewers for their insightful and constructive comments.
> - (W1) Aggregating 13 datasets raises questions about licenses and usage permissions. Please include a table listing the license, citation, and redistribution policy for each source.
>
> In the revised manuscript, we have added Table 5, which now includes (i) the license type, citation, and redistribution policy for each dataset, and (ii) additional details on scanner vendors, gestational age ranges, and geographic distribution to support generalizability.
>
> - (W2) Converting masks to boxes and asking “what’s in the box?” evaluates recognition, not localization fidelity. Consider adding a pointing game (models output coordinates of the bounding box).
>
> We agree that predicting coordinates would provide a better assessment of localization fidelity. However, since VLMs are not well-trained to handle numerical outputs and are known to struggle with numbers, we adopt a more intuitive multiple-choice (MCQ) setup instead. The bounding box is provided within the image, and the model is asked to identify what is inside. Even with this simplified formulation, most models still fail to select the correct answer, highlighting the challenge.
>
> - (W3) “normal/benign/malignant” in fetal ultrasound is non-trivial; please clarify labeling sources, consensus rules, and whether images are frame-level vs case-level labels.
>
> All images are sourced from a public dataset comprising exclusively fetal head ultrasound frames. Each image is annotated at the frame level with one of three diagnostic categories: normal, benign, or malignant. We rely on the available frame-level labels provided with the dataset. We further clarified this in the paper.
>
> - (W4) Several source datasets include video sweeps; current tasks are frame-centric. Adding temporal subtasks would align with real ultrasound operation and may favor models with spatiotemporal reasoning.
>
> We agree that temporal reasoning is crucial for accurate and realistic ultrasound interpretation. However, most fetal ultrasound analysis tasks are frame-based, while video-based studies mainly target applications such as cardiac or flow analysis. As no suitable public video datasets were available for generating question–answer pairs, we focused on frame-centric tasks, which remain the standard setting in this domain.
>
> - (W5) Multiple typos. Section 4.4 seems to be incomplete.
>
> We thank the reviewer for bringing this to our attention. We have corrected the typos and fixed the formatting issue in Section 4.4. The complete text now reads:
> “After task-specific fine-tuning, Llama-3.2-11B improves substantially from 33% to 85% overall accuracy, with consistent gains across all tasks. Qwen2.5-VL also shows clear improvements, increasing from 24% to 52% overall accuracy across the same tasks.”
>
> - (Q1) Is a single-frame image enough for FO or CD tasks? Sonographers may rely on multi-view reasoning, temporal sweep, and 3D relationships.
>
> While sonographers typically leverage multi-view reasoning and temporal sweeps, our results demonstrate that a single-frame image can still provide sufficient diagnostic information for both FO and CD tasks. For Clinical Diagnosis (CD) involving fetal head conditions, the three standard views (transventricular, transthalamic, and transcerebellar) are recommended because they capture most fetal head abnormalities [1]. Our model, trained on these representative planes, effectively learns the discriminative features necessary for CD even from a single frame. Similarly, for Fetal Orientation (FO), although sonographers often rely on two images [2], our model achieves 85% accuracy from a single frame, indicating strong spatial reasoning capability and suggesting that one image often contains sufficient orientation cues.
>
> [1] Brétagnolle V, Salaün P, Bazot M, Achiron R. Normal and abnormal fetal brain development during the third trimester. Eur J Radiol. 2005;57(2):177-94. doi:10.1016/S0720-048X(05)00387-6.
>
> [2] Bronshtein M, Gover A, Zimmer EZ. Sonographic definition of the fetal situs. Obstet Gynecol. 2002 Jun;99(6):1129-30. doi: 10.1016/s0029-7844(02)02017-3. PMID: 12052611.

---

> > ### Author Response · Authors · 2025-11-27
> >
> > We hope our clarifications have addressed your concerns and demonstrated the robustness of our contributions. If any additional details would help, we are happy to provide them.

---

### Meta-Review · Area_Chair_wQxd · 2025-12-18

**Summary:**

This work introduces a benchmark for Vision-Language Models in fetal ultrasound, with 42k images and 93k QA pairs. Three reviewers have reviewed this paper, with two marginally above the acceptance threshold (rating 6) and one marginally below the acceptance threshold (rating 4). Reviewer 4LjX initially gave a rating of 4 and agreed to improve the rating after the response.

Reviewer aVZ6 raised concerns about the details of datasets and tasks. The authors have responded with detailed revisions. No significant concerns remained.

Reviewer ghap was concerned about the IRB approval, while the manuscript showed that all datasets are publicly available. For clinical evaluation, the authors said that the current models still struggle with unsatisfactory results. I can accept it since the main contribution lies in the introduced benchmark. In my view, the raised questions have been well addressed.

Reviewer 4LjX was mainly concerned about the processing details of datasets. I think the original concerns are not very significant, and the authors have carefully responded to the raised concerns.

Based on the positive rating and the authors’ response, I am glad to recommend acceptance.

**Reviewer Concerns:**

No significant concerns remain.

**Reviewer Scores:**

Reviewer 4LjX may improve the rating.

---

### Decision · Program_Chairs · 2026-01-26

Accept (Poster)